# Predictive Model of Good Clinical Outcomes in Patients Undergoing Coronary Angiography after Out-of-Hospital Cardiac Arrest: A Prospective, Multicenter Observational Study Conducted by the Korean Cardiac Arrest Research Consortium

**DOI:** 10.3390/jcm10163695

**Published:** 2021-08-20

**Authors:** Jin Ho Beom, Incheol Park, Je Sung You, Yun Ho Roh, Min Joung Kim, Yoo Seok Park

**Affiliations:** 1Department of Emergency Medicine, Yonsei University College of Medicine, Seoul 03722, Korea; wangtiger@yuhs.ac (J.H.B.); incheol@yuhs.ac (I.P.); youjsmed@yuhs.ac (J.S.Y.); 2Biostatistics Collaboration Unit, Department of Biomedical Systems Informatics, Yonsei University College of Medicine, Seoul 03722, Korea; yunhoroh@yuhs.ac

**Keywords:** nomograms, out-of-hospital cardiac arrest, coronary angiography, emergency medical technicians, return of spontaneous circulation, emergency medical services, electrocardiography

## Abstract

This observational study aimed to develop novel nomograms that predict the benefits of coronary angiography (CAG) after resuscitating patients with out-of-hospital cardiac arrest (OHCA) regardless of the electrocardiography findings and to perform an external validation of these models. Data were extracted from a prospective, multicenter registry of resuscitated patients with OHCA (October 2015–June 2018). New nomograms were developed based on variables associated with survival discharge and neurologic outcomes; their analysis included 723 and 709 patients, respectively. Patient age (*p* < 0.001), prehospital defibrillation by emergency medical technicians (EMTs) (*p* = 0.003), prehospital return of spontaneous circulation (ROSC) (*p* = 0.02), and time from collapse to ROSC (*p* < 0.001) were associated with survival discharge. Patient age (*p* < 0.001), prehospital defibrillation by EMTs (*p* < 0.001), and time from collapse to ROSC (*p* < 0.001) were associated with neurologic outcomes. The new nomogram had a good predictive performance, with an area under the curve (AUC) of 0.8832 (95% confidence interval (CI): 0.8358–0.9305) for survival discharge and an AUC of 0.9048 (95% CI: 0.8627–0.9469) for neurologic outcomes. Novel nomograms that predict survival discharge and good neurological outcomes after CAG in patients with OHCA were developed and validated; they can be quickly and easily applied to identify patients who will benefit from CAG.

## 1. Introduction

The management of out-of-hospital cardiac arrest (OHCA) remains a challenge worldwide [1,2]. The current European Resuscitation Council and American Heart Association guidelines recommend that all patients with cardiac arrest with a suspected cardiac cause of arrest and ST-segment elevation on electrocardiography (ECG) undergo emergent coronary angiography (CAG) [3,4]. In addition, these guidelines state that emergent CAG is reasonable for some adult patients with OHCA and suspected cardiac origin who do not have ST-segment elevation on ECG (such as those who are electrically or hemodynamically unstable) [3,4].

Several previous observational studies and meta-analyses including patients without ST-segment elevation on ECG reported a correlation between CAG and improved neurological outcomes [5,6,7]. However, the Coronary Angiography after Cardiac Arrest without ST Segment Elevation trial, conducted by Lemkes et al., reported no improvement in the survival rates of patients resuscitated after OHCA with an initial shockable rhythm in whom no ST segment elevation or signs of shock were present. Additionally, while coronary artery disease was found in 65% of patients who underwent CAG, most of whom were in a stable state, only approximately 15% of the patients had acute unstable lesions, and thrombotic occlusion was reported in only 5% of them [8]. Therefore, not all patients who undergo successful resuscitation after cardiac arrest benefit from CAG. 

The objective of this study was to develop novel nomograms that can predict the prognosis of patients who undergo CAG regardless of the ECG findings after OHCA and to perform an external validation of the novel nomograms. The nomograms developed in this study provide a visual representation of a statistical predictive model that produces numerical probabilities of clinical events, which is more exact than the conventional method that uses odds ratios to predict prognosis [9].

## 2. Materials and Methods

### 2.1. Study Design and Setting

This observational study utilized data from a prospective, multicenter registry (Korean Cardiac Arrest Research Consortium (KoCARC) registry) of patients with OHCA who were resuscitated between October 2015 and June 2018. The patients received cardiopulmonary resuscitation (CPR) in accordance with advanced cardiac life support recommendations; targeted temperature management (TTM), extracorporeal CPR (E-CPR), CAG, and percutaneous coronary intervention (PCI) were conducted according to each institution’s protocol. The KoCARC is a multicenter, nationwide collaborative research network of 62 participating secondary or tertiary hospitals that was established to comprehend the multitude of studies conducted in the field of OHCA and to reinforce the cooperative effort in conducting these studies [10]. The KoCARC registry was designed to continuously enroll patients with non-traumatic OHCA, who were transported to the emergency departments (EDs) of the participating hospitals. This study excluded patients with OHCA with obvious non-cardiac etiologies, such as trauma, drowning, poisoning, burns, asphyxia, or hanging. In addition, the registry excluded patients with OHCA who had terminal illnesses documented by medical records, patients under hospice care, pregnant women, and patients with pre-documented do-not-attempt-resuscitation cards [10]. This study was approved by the Institutional Review Board of Yonsei University College of Medicine, Severance Hospital (no. 4-2015-1162) as well as the institutional review boards of each of the participating hospitals. The KoCARC registry database was registered at clinicaltrials.gov as protocol number NCT03222999.

### 2.2. Korean Emergency Medical Service (EMS) System

The Korean EMS system is a basic-to-intermediate level ambulance service operated by 16 provincial headquarters of the National Fire Department and a single-tiered, fire-based EMS system [11]. The most qualified emergency medical technician (EMT) performs CPR with an automated external defibrillator, evaluates the cardiac rhythm at scene, provides advanced airway management, and administers intravenous fluids. EMTs are not allowed to declare death or stop CPR on the scene unless the patient regains a pulse in the field or during transport to an ED; therefore, all EMS-assessed patients are transported to the nearest ED. In the case of prehospital EMS in Korea, E-CPR is not implemented [12].

### 2.3. Study Population

The KoCARC registry included data of 874 patients, who underwent CAG regardless of ECG findings (both ST elevation and non-ST elevation) after the return of spontaneous circulation (ROSC), registered from October 2015 to June 2018. In this study, patients aged <18 years, those transferred from other hospitals after ROSC, and those with unknown primary or secondary outcomes at hospital discharge were excluded. For time-related variables (including the time required for emergency medical service (EMS) response), values that were more than two standard deviations (SDs) from the mean were excluded from the data analysis [13].

### 2.4. Data Collection

All data of the KoCARC registry were anonymized and collected using a web-based case report form (CRF) by the research coordinator at each institution. The CRF comprised seven research fields classified with variables related to OHCA. Each field contained core and optional variables. To improve data quality, the web-based data entry system filtered out outliers or inaccurate values. The research coordinator of each participating hospital was responsible for ensuring data accuracy. The research committee sought to improve the quality of the data by regularly monitoring the data and providing continuous feedback regarding inaccurate data [10]. 

The following data were retrieved from the KoCARC registry: patient demographics, comorbidities (hypertension, diabetes mellitus, and dyslipidemia), prehospital characteristics (witnessed arrest, place of arrest, bystander cardiopulmonary resuscitation [CPR], prehospital defibrillation by bystanders or EMS personnel, primary ECG rhythm at the scene, and epinephrine use by EMS providers), hospital characteristics (ROSC at ED arrival, initial ECG rhythm at hospital, and total dose of epinephrine used at hospital), post-ROSC characteristics (E-CPR, TTM, coronary angiographic finding, PCI, and vasopressor use in the hospital), cardiac markers, and time intervals (time from EMS call to scene arrival, time from collapse to ED arrival, time from collapse to ROSC, and time from ED arrival to CAG). 

### 2.5. Outcome Measurements

The primary endpoint was survival at hospital discharge. The secondary outcome was a favorable neurologic recovery at hospital discharge, defined as a cerebral performance category (CPC) score of 1 or 2.

### 2.6. Statistical Analyses

All statistical analyses were performed using the R software, version 3.4.3 for Windows (The R Foundation for Statistical Computing, Vienna, Austria). Continuous variables are presented as means ± SDs, and categorical variables are presented as frequencies (percentages). Independent two-sample t-tests were used to compare continuous variables, and the chi-squared test or Fisher’s exact test was used to compare categorical variables, as appropriate.

The study population was randomly allocated to either a training or validation set at a 7:3 ratio using the split sample validation method. In the training set, a univariate logistic regression analysis was performed to evaluate the prognostic ability of each variable for the primary and secondary outcomes. Then, a multivariable logistic regression analysis with variables with <20% missing data that had a *p*-value < 0.05 on the univariate binary logistic regression was performed. The variables that were identified as independently associated with survival or neurologic outcomes at hospital discharge were used to create the novel nomograms using the training data set. The performances of the nomograms were evaluated with respect to discrimination and calibration [14]. The predictive accuracies (discriminations) of the models were assessed using area under the receiver-operating characteristic curve (AUC) values and 95% confidence intervals (CIs), which quantify the level of concordance between the predicted probabilities and actual chance of the event of interest occurring. The accuracy of the novel nomograms was internally validated using the bootstrap method using resampling 1000 times of simple random sampling with replacement. The calibration of the nomograms represents how accurate the predicted probabilities are compared with the observed outcome frequencies using graphic representations (calibration curves). A curve along a 45-degree line indicates a perfect calibration model in which the predicted probabilities are identical to the actual outcomes. The calibration curves are presented as apparent and bias-corrected calibration plots using the bootstrapping methods with 500 re-samples. The predictive accuracy and calibration of the novel nomograms generated with the training set were subsequently tested in the validation set. In addition, to verify whether our novel nomogram would be applicable to the patients irrespective of the CAG findings, we performed the sensitivity analysis. According to the CAG findings, the group was divided into patients without actual lesions (normal coronary, insignificant lesion) and patients with actual lesions or decreased blood flow (notable stenosis, vasospasm). All reported *p*-values are two-sided, and statistical significance was set at *p* < 0.05.

## 3. Results

Within the study period, 7576 patients with OHCA were registered in the KoCARC registry. Of these, 874 patients underwent CAG after ROSC. Patients who transferred from other hospitals (*n* = 108), were aged <18 years (*n* = 1), or had missing data regarding mortality (*n* = 17) or CPC score (*n* = 14) at hospital discharge and patients with data outliers of time variables (*n* = 25) were excluded from the study. The final analyses included 723 patients for the survival outcome group and 709 patients for the neurological outcome group (Figure 1). Among the 723 patients in the survival outcome group, we could confirm the CAG findings in 519 patients in the KoCARC registry: normal coronary findings in 84 patients (16.2%), an insignificant lesion in 69 patients (13.3%), significant stenosis in 303 patients (58.4%), and vasospasm in 63 patients (12.1%). Of the 723 patients in the survival outcome group, 273 (168 survivors/105 deaths) underwent PCI. Among the 709 patients in the neurological outcome group, we could confirm the CAG findings in 508 patients: normal coronary, 83 patients (16.3%); insignificant lesion, 67 patients (13.2%); significant stenosis, 296 patients (58.3%); and vasospasm, 62 patients (12.2%). Among the 709 patients in the neurological outcome group, 268 (144 good neurologic outcomes/124 poor neurologic outcomes) underwent PCI.

The training set for the survival outcome analyses included 496 patients, and the validation set included 227 patients. The baseline characteristics are summarized in Table 1. 

In the univariate logistic regression analysis, patient age, medical history, bystander CPR, shockable rhythm at the scene, prehospital defibrillation by bystanders or EMS providers, epinephrine use by EMS providers, prehospital ROSC, initial shockable rhythm at hospital arrival, total dose of epinephrine used during CPR in the hospital, diastolic blood pressure, vasopressor use in the hospital, and time-related variables (time from EMS call to scene arrival, time from collapse to ROSC, and time from ED arrival to CAG) were identified as significantly associated with survival discharge after CAG (Table 1). Among them, patient age (*p* < 0.001), prehospital defibrillation by EMS providers (*p* = 0.003), prehospital ROSC (*p* = 0.02), and time from collapse to ROSC (*p* < 0.001) were identified as independent prognostic factors for survival discharge using multivariable logistic regression analysis (Table 2). 

These four prognostic factors were included in the novel nomogram for predicting survival discharge after CAG. The time from collapse to ROSC (min) was the factor that contributed the most to the prediction of survival discharge after performing CAG, followed by patient age (Figure 2a). The total points of the variables in the nomogram allowed for the prediction of the survival discharge probability of the patients in the CAG group. For example, a 65-year-old (25 points) patient who was defibrillated by EMS providers (20 points) with ROSC before hospital arrival (17.5 points) and a time from collapse to ROSC of 10 min (92.5 points) would have a total score of 155 and a predicted probability of survival discharge of approximately 90%. The discriminative ability of the novel nomogram was good, with an AUC of 0.8832 (95% CI: 0.8358–0.9305). The AUC of the internally validated nomogram was 0.8881 (95% CI: 0.8386–0.9339), with a statistical power similar to that of the initial nomogram. In the validation set, the discrimination was excellent, with an AUC of 0.9002 (95% CI: 0.869–0.9315) (Figure 2b). The calibration plot of the nomogram demonstrated an excellent agreement between the predicted and observed probabilities of survival discharge in both sets (Figure 2c). The sensitivity analysis in patients with or without actual lesions showed that the AUC values of the novel nomogram were 0.8557 (95% CI: 0.7832–0.9281) and 0.9439 (95% CI: 0.8824–0.9999), which were not significantly different from that of the initial nomogram (*p* = 0.53 and *p* = 0.13, respectively).

The training set for the neurologic outcome analyses included 489 patients, and the validation set included 220 patients. The baseline characteristics and variables found to be associated with favorable neurologic outcomes are summarized in Table 3. Patient age (*p* < 0.001), prehospital defibrillation by EMS providers (*p* < 0.001), and time from collapse to ROSC (*p* < 0.001) were identified as independent prognostic factors for neurologic outcomes (Table 2). 

These three prognostic factors were included in the novel nomogram for predicting neurologic outcomes at hospital discharge. The time from collapse to ROSC was the most contributing factor in this nomogram (Figure 3a). The AUC of the nomogram was 0.9048 (95% CI: 0.8627–0.9469), and the internally validated AUC was 0.9070 (95% CI: 0.8554–0.9509). In the validation set, the discrimination of the nomogram was good, with an AUC of 0.8742 (95% CI: 0.8383–0.9101) (Figure 3b). The calibration plot of the nomogram demonstrated a good agreement in both sets (Figure 3c). The sensitivity analysis in patients with or without actual lesions showed that the AUC values of the novel nomogram were 0.8906 (95% CI: 0.8234–0.9579) and 0.9490 (95% CI: 0.8872–0.9999), which were not significantly different to that of the initial nomogram (*p* = 0.73 and *p* = 0.25, respectively).

## 4. Discussion

In this study, novel nomograms that predict survival discharge and good neurological outcomes for performing CAG in patients with OHCA were developed and validated. To the best of our knowledge, this is the first study to create a nomogram that determines if CAG will be beneficial by considering various variables in patients with OHCA. These nomograms were based on patient age, prehospital ROSC, prehospital defibrillation by EMS providers, and time from collapse to ROSC, which are readily available data at the prehospital and in-hospital stages of treatment. Therefore, they can be used in an emergency situation and easily applied to each patient individually.

The time from collapse to ROSC was the most contributing variable in the nomograms predicting survival discharge and good neurologic outcomes at hospital discharge after CAG in patients with OHCA. The time from collapse to ROSC may be associated with ischemia. Cardiac ischemia and adenosine triphosphate depletion are believed to worsen over time after cardiac arrest [15]. Longer ischemic periods result in worsened neurological prognoses, and ultimately, increased probabilities of death. Therefore, the time from collapse to ROSC affects the severity of hypoxic-ischemic brain injuries and is one of the major causes of death after cardiac arrest [16,17]. Thus, the time from collapse to ROSC is an important factor for determining the benefits of CAG. Patient age and prehospital defibrillation by EMS providers are also common variables to both nomograms. Several previous studies have reported that a younger age and the use of automated external defibrillators at the scene are associated with better outcomes for patients with OHCA [18,19,20]. Patient age has more influence on the survival discharge in this study. Prehospital ROSC was included on the survival discharge nomogram and not on the neurologic outcome nomogram. A study conducted by David et al. reported that cardiac arrest survival is rare without prehospital ROSC [21]. Another study reported that patients with an initial shockable rhythm or EMS-witnessed arrest in the absence of field ROSC had a survival rate of 0.5% [22]. The results of these previous studies support the nomograms developed in this study.

Recent studies have reported that performing CAG early improved the neurological outcomes and survival rate regardless of ECG findings in survivors of OHCA [23,24]. One study showed that early CAG may be associated with improved survival in patients with bystander-witnessed OHCA with shockable rhythms [25]. However, other studies have reported no association between early CAG and survival in patients with non-ST elevation after cardiac arrest [26,27]. In this study, the time from ED arrival to CAG was approximately 10 times longer in the survival group than in the deceased group (2042.5 ± 4903.8 min vs. 204.6 ± 727.3 min; *p* < 0.001) and approximately five times longer in the good neurologic outcome group than in the poor neurologic outcome group (2184.0 ± 4916.1 min vs. 483.7 ± 2535.2 min; *p* < 0.001). These results should not be interpreted to mean that delayed or late CAG results in better outcomes than early CAG. Rather, these results may be attributed to the fact that severely ill patients often receive early CAG, resulting in decreased survival rates and poor neurological outcomes among patients who receive early CAG. In contrast, hemodynamically stable patients who have regained consciousness may receive delayed CAG. The results of previous studies combined with those of this study indicate that it is difficult to decide whether to perform CAG simply based on time because of the various clinical situations that act as confounding factors. Therefore, the novel nomogram developed in this study is a good alternative for deciding whether or not to perform CAG.

This study is not without limitations. First, despite the constant efforts of the KoCARC registry committee to improve the reliability and accuracy of the data, some patients had inaccurate or missing data, which could not be included in this study. Second, only data regarding survival and neurologic prognoses at the time of hospital discharge were collected, although long-term data were not collected in this study. Third, some outlier data for time-related variables were excluded to increase the reliability of the study. Fourth, this study did not include the data of some variables such as echocardiographic finding, which was important to identify clinically relevant characteristics of the patients. Fifth, in the case of patients who underwent E-CPR, different results were obtained, but the analysis was not conducted owing to the small number of patients who underwent E-CPR in this study. Finally, data regarding ST segment elevation on the initial ECG were not available in the KoCARC registry. For the practical application of the newly developed nomogram, further validations including patients with suspected cardiac causes of arrest and non-ST segment elevation are needed.

## 5. Conclusions

Novel nomograms that predict survival discharge and good neurological outcomes of patients with OHCA undergoing CAG were developed and validated in this study. These nomograms use data that can be obtained during the prehospital and hospital resuscitation period and can be quickly and easily applied to decide if CAG should be performed in resuscitated patients with OHCA.

## Figures and Tables

**Figure 1 jcm-10-03695-f001:**
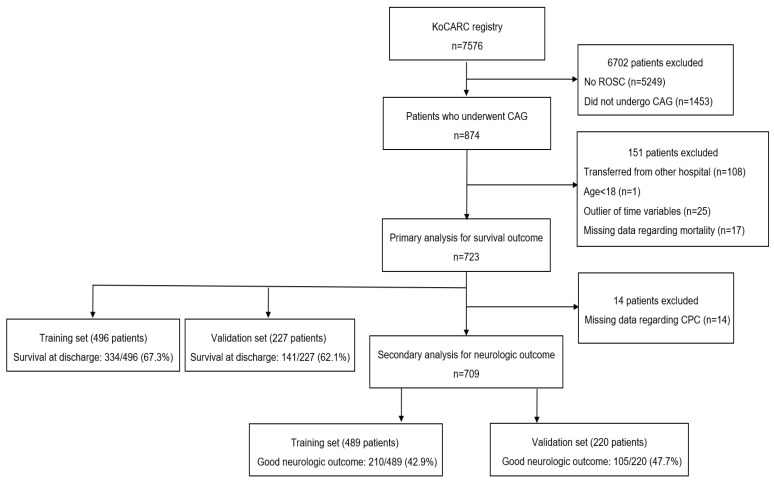
Patient flowchart. Abbreviations: KoCARC: Korean Cardiac Arrest Research Consortium, ROSC: return of spontaneous circulation, CAG: coronary angiography, CPC: cerebral performance category.

**Figure 2 jcm-10-03695-f002:**
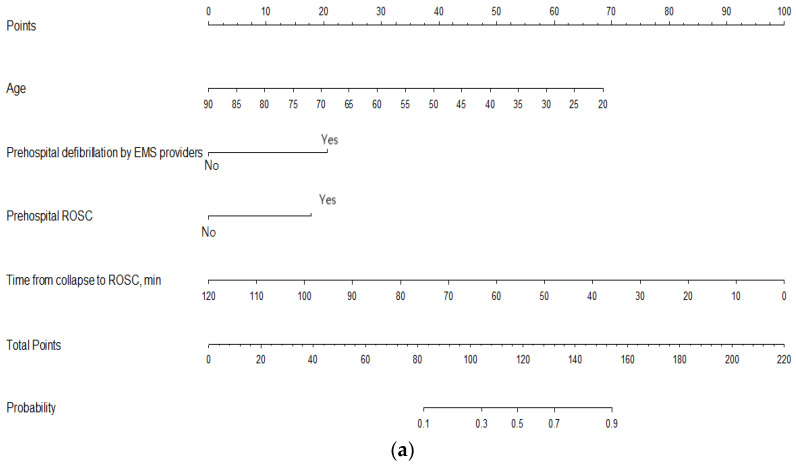
Nomogram for predicting survival discharge. (**a**) The novel nomogram for predicting survival discharge developed using the training data set is shown. (**b**) The discriminative ability of the nomogram is excellent, with an area under the curve of 0.9002 (95% confidence interval: 0.869–0.9315). (**c**) The calibration plot of the prediction model indicates good agreement between the predicted and observed probabilities of survival discharge. Abbreviations: EMS: emergency medical service, ROSC: return of spontaneous circulation.

**Figure 3 jcm-10-03695-f003:**
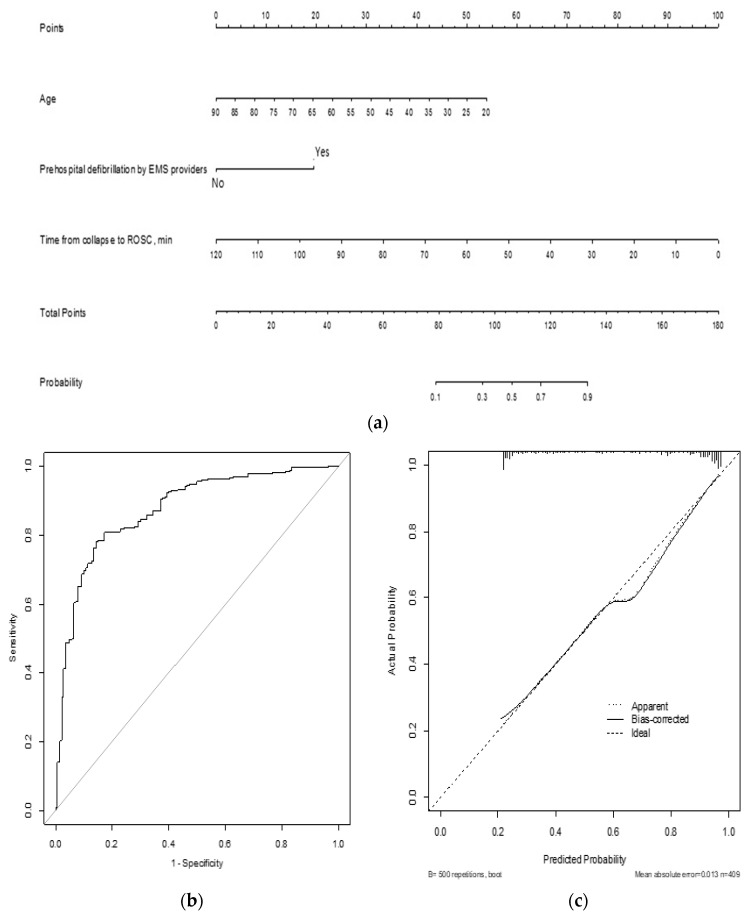
Nomogram for predicting good neurologic outcomes. (**a**) The novel nomogram for predicting good neurologic outcomes developed using the training data set is shown. (**b**) The discriminative ability of the nomogram is good, with an area under the curve of 0.8742 (95% confidence interval: 0.8383–0.9101). (**c**) The calibration plot of the prediction model indicates good agreement between the predicted and observed probabilities of good neurologic outcomes. Abbreviations: EMS: emergency medical service, ROSC: return of spontaneous circulation.

**Table 1 jcm-10-03695-t001:** Patient demographics and clinical characteristics according to survival outcome at discharge.

	Training Set (*n* = 496)			Validation Set (*n* = 227)		
	Survival (*n* = 334)	Death (*n* = 162)	*p*-Value	Survival (*n* = 141)	Death (*n* = 86)	*p*-Value
Age (years)	55.3 ± 13.0	64.3 ± 13.3	<0.001	56.1 ± 13.1	63.3 ± 13.8	<0.001
Sex, male	279 (83.5)	133 (82.1)	0.69	123 (87.2)	72 (83.7)	0.46
Medical history						
Hypertension	122 (39.1)	80 (52.0)	0.008	64 (46.7)	39 (47.6)	0.9
Diabetes mellitus	65 (21.0)	55 (36.2)	<0.001	24 (17.8)	29 (35.4)	0.003
Dyslipidemia	23 (7.80)	9 (6.2)	0.53	11 (8.6)	7 (9.6)	0.81
Bystander-witnessed	275 (84.1)	125 (77.2)	0.06	108 (77.1)	66 (76.7)	0.95
Place of arrest			0.02			0.09
Public	185 (44.8)	75 (46.9)		74 (53.6)	41 (47.6)	
Private	122 (37.7)	76 (47.5)		60 (43.5)	36 (41.9)	
Ambulance	16 (4.9)	9 (5.6)		4 (2.9)	9 (10.5)	
Bystander CPR			0.008			0.1
Chest compressions	193 (61.3)	73 (46.8)		83 (62.4)	42 (50.0)	
Prehospital defibrillation by bystanders	14 (4.4)	1 (0.6)	0.03	6 (4.6)	1 (1.2)	0.18
Primary cardiac rhythm at the scene			<0.001			<0.001
Shockable	263 (85.1)	73 (46.5)		113 (85.0)	42 (51.9)	
Prehospital defibrillation by EMS providers	285 (85.6)	81 (51.3)	<0.001	119 (85.6)	45 (52.3)	<0.001
Epinephrine use by EMS providers	24 (7.2)	21 (13.0)	0.04	16 (11.4)	15 (17.4)	0.2
Prehospital ROSC, yes	241 (72.2)	20 (12.4)	<0.001	101 (71.6)	11 (12.8)	<0.001
Primary cardiac rhythm at hospital			<0.001			0.27
Shockable	36 (45.6)	24 (17.52)		11 (35.5)	20 (28.7)	
Full dose of epinephrine used during CPR in the hospital (mg)	1.6 ± 3.3	7.1 ± 6.6	<0.001	1.5± 2.9	5.7 ± 5.4	<0.001
SBP (mmHg)	129.0 ± 39.5	117.5 ± 40.8	0.07	133.2 ± 38.5	121.3 ± 41.7	0.17
DBP (mmHg)	82.4 ± 22.0	73.4 ± 28.1	0.02	83.5± 26.5	79.8 ± 32.4	0.55
HR (beats/min)	103.1 ± 30.6	98.2 ±32.0	0.32	100.2 ± 31.9	93.3 ± 34.1	0.34
Troponin I	74.4 ± 650.3	24.5 ± 141.9	0.42	32.4 ± 254.4	8.5 ± 26.7	0.45
E-CPR	18 (6.2)	48 (29.3)	<0.001	6 (4.6)	20 (28.2)	<0.001
Targeted temperature management	132 (40.7)	52 (32.7)	0.17	44 (32.8)	28 (33.3)	0.61
PCI	99 (35.1)	66 (41.0)	0.22	43 (32.1)	30 (46.2)	0.05
Vasopressor use in the hospital	177 (54.3)	149 (92.0)	<0.001	82 (60.3)	78 (91.8)	<0.001
Time from EMS call to scene arrival, min	7.1 ± 3.6	8.2 ± 3.6	0.002	7.9 ± 4.7	8.6 ± 5.3	0.33
Time from collapse to ED arrival, min	30.3 ± 13.2	30.4 ± 15.2	0.93	31.3 ± 14.1	30.0 ± 13.0	0.51
Time from collapse to ROSC, min	21.0 ± 14.1	44.3 ± 25.2	<0.001	21.9 ± 17.0	44.7 ± 24.0	<0.001
Time from ED arrival to CAG, min	2042.5 ± 4903.8	204.6 ± 727.3	<0.001	1781.4± 5702.3	978.9 ± 5196.9	0.29

Variables are shown as means ± standard deviations or numbers (percentages). Abbreviations: CPR: cardiopulmonary resuscitation, EMS: emergency medical service, ROSC: return of spontaneous circulation, SBP: systolic blood pressure, DBP: diastolic blood pressure, HR: heart rate, E-CPR: extracorporeal cardiopulmonary resuscitation, PCI: percutaneous coronary intervention, ED: emergency department, CAG: coronary angiography.

**Table 2 jcm-10-03695-t002:** Predictors of survival discharge and good neurologic outcomes.

	Survival Discharge		Good Neurologic Outcomes	
	OR (95% CI)	*p*-Value	OR (95% CI)	*p*-Value
Age	0.94 (0.90–0.98)	<0.001	0.95 (0.92–0.98)	<0.001
Prehospital defibrillation by EMS providers	3.52 (1.53–8.05)	0.003	7.84 (3.05–20.17)	<0.001
Prehospital ROSC	2.96 (1.20–7.29)	0.02		
Time from collapse to ROSC, min	0.95 (0.93–0.97)	<0.001	0.91 (0.88–0.94)	<0.001

Abbreviations: OR: odds radio, CI: confidence interval, EMS: emergency medical service, ROSC: return of spontaneous circulation.

**Table 3 jcm-10-03695-t003:** Patient demographics and clinical characteristics according to neurologic outcomes at discharge.

Variables	Training Set (*n* = 489)			Validation Set (*n* = 220)		
	Good Neurologic Outcome (CPC 1, 2) (*n* = 210)	Poor Neurologic Outcome (CPC ≥ 3) (*n* = 279)	*p*-Value	Good Neurologic Outcome (CPC 1, 2) (*n* = 105)	Poor Neurologic Outcome (CPC ≥ 3) (*n* = 115)	*p*-Value
Age (years)	54.6 ± 12.7	63.0 ± 13.6	<0.001	55.0 ± 12.8	62.7 ± 13.5	<0.001
Sex, male	234 (83.9)	170 (80.9)	0.4	100 (86.9)	90 (85.7)	0.79
Medical history						
Hypertension	101 (38.4)	99 (50.3)	0.01	52 (46.9)	47 (46.1)	0.91
Diabetes mellitus	47 (18.1)	71 (36.4)	<0.001	17 (15.6)	35 (34.7)	0.001
Dyslipidemia	22 (8.76)	10 (5.41)	0.18	11 (10.7)	7 (7.5)	0.45
Bystander-witnessed	232 (84.7)	162 (77.9)	0.06	87 (76.3)	82 (78.1)	0.75
Place of arrest			0.02			0.003
Public	157 (57.7)	104 (50.5)		58 (51.8)	54 (51.5)	
Private	103 (37.9)	89 (43.2)		51 (45.5)	41 (39.1)	
Ambulance	12 (4.4)	13 (6.3)		3 (2.7)	10 (9.5)	
Bystander CPR			<0.001			0.01
Chest compression	174 (65.4)	91 (45.5)		73 (66.4)	50 (49.5)	
Prehospital defibrillation by bystanders	13 (4.9)	2 (1.0)	0.02	5 (4.6)	2 (2.0)	0.31
Primary cardiac rhythm at the scene			<0.001			<0.001
Shockable	233 (90.3)	99 (48.7)		100 (92.6)	53 (53.00)	
Prehospital defibrillation by EMS providers	250 (89.9)	111 (53.6)	<0.001	103 (91.2)	58 (55.2)	<0.001
Epinephrine use by EMS providers	16 (5.7)	29 (13.8)	0.002	12 (10.4)	18 (17.1)	0.15
Prehospital ROSC, yes	226 (81.0)	34 (16.2)	<0.001	93 (80.9)	18 (17.1)	<0.001
Primary cardiac rhythm at hospital			<0.001			0.005
Shockable	23 (53.5)	37 (21.9)		10 (71.4)	20 (24.1)	
Full dose of epinephrine used during CPR in the hospital (mg)	1.0 ± 2.8	6.2 ± 6.1	<0.001	1.2 ± 2.9	5.0 ± 5.2	<0.001
SBP (mmHg)	129.4 ± 36.4	122.0 ± 44.8	0.19	133.7 ± 39.3	122.9 ± 41.1	0.2
DBP (mmHg)	83.7± 20.5	74.7 ± 27.8	0.01	85.2 ± 27.9	78.6 ± 29.1	0.28
HR (beats/min)	101.8 ± 28.7	102.6 ± 34.1	0.86	99.3 ± 29.7	95.7 ± 35.7	0.61
Troponin I	85.8 ± 705.6	21.6 ± 127.6	0.28	39.8 ± 283.6	7.4 ± 24.4	0.31
E-CPR	9 (3.9)	55 (26.1)	<0.001	3 (2.7)	22 (25.0)	<0.001
Targeted temperature management	103 (38.0)	82 (40.0)	0.63	37 (33.6)	34 (33.3)	0.96
PCI	85 (36.8)	76 (36.9)	0.98	35 (30.4)	38 (45.8)	0.03
Vasopressor use in the hospital	137 (50.4)	184 (88.0)	<0.001	59 (53.6)	95 (90.5)	<0.001
Time from EMS call to scene arrival, min	6.9 ± 3.3	8.1 ± 3.9	<0.001	7.6 ± 4.9	8.8 ± 5.0	0.08
Time from collapse to ED arrival, min	30.2 ± 13.4	30.5 ± 14.7	0.84	30.9 ± 14.3	30.7 ± 13.3	0.91
Time from collapse to ROSC, min	18.3 ± 11.7	42.0 ± 24.1	<0.001	19.6 ± 16.5	42.6 ± 22.9	<0.001
Time from ED arrival to CAG, min	2184.0 ± 4916.1	483.7 ± 2535.2	<0.001	2218.1 ± 6287.7	828.4 ± 4676.9	0.07

Variables are shown as means ± standard deviations or numbers (percentages). Abbreviations: CPC: cerebral performance category, CPR: cardiopulmonary resuscitation, EMS: emergency medical service, ROSC: return of spontaneous circulation, SBP: systolic blood pressure, DBP: diastolic blood pressure, HR: heart rate, E-CPR: extracorporeal cardiopulmonary resuscitation, PCI: percutaneous coronary intervention, ED: emergency department, CAG: coronary angiography.

## Data Availability

Restrictions apply to the availability of these data. Data was obtained from KoCARC and are available with the permission of KoCARC.

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
