# Peer review of "Predictive Model of Good Clinical Outcomes in Patients Undergoing Coronary Angiography after Out-of-Hospital Cardiac Arrest: A Prospective, Multicenter Observational Study Conducted by the Korean Cardiac Arrest Research Consortium"

_jcm, 2021, doi:10.3390/jcm10163695_

Round 1
Reviewer 1 Report
Beom et al. conducted a retrospective analysis of Korean registry data for out-of-hospital cardiac arrest patients. The aim of the analysis is to develop a predictive score for the prognosis after coronary angiography, regardless of ECG findings. To address this question, the research team analyzed 723 patients for survival to hospital discharge and 709 patients for the neurological outcomes. The authors concluded that age, prehospital defibrillation, return of spontaneous circulation, and time to collapse were associated with survival to discharge. Also, age, prehospital defibrillation, and time to collapse were associated with good neurological outcomes.
The study is of moderate quality and has a good sample size. Also, the outcomes are good and need to be reported.
The major concern I have about this study is the study design does not address the research question. All the current design targets the outcome either to hospital discharge or the neurological recovery. The determining factors are confounders for the need for a heart catheterization. However, there are no reported data on the cath findings, and how many are false positives or negatives. Also, we need data related to cardiac markers, echo..etc.
In addition, I wish the authors shed light on the clinical protocols for E-CPR. Is it standardized among the institutions? EMS protocol pre-hospital and heart cath after ROSC.
Author Response
Responses to Reviewer #1
The major concern I have about this study is the study design does not address the research question. All the current design targets the outcome either to hospital discharge or the neurological recovery. The determining factors are confounders for the need for a heart catheterization. However, there are no reported data on the cath findings, and how many are false positives or negatives
Response: Thank you for taking the time to review our manuscript and for providing insightful comments. Our study aimed to predict the prognosis of patients undergoing coronary angiography, not to identify patients with actual lesions. Therefore, the CAG result was not included as a confounding variable. However, we agree that it is important to report CAG results in our study. As suggested, we have added the CAG results in the Results section as shown below:
“Among the 723 patients in the survival outcome group, we could confirm the CAG findings in 519 patients in the KoCARC registry: normal coronary findings in 84 patients, an insignificant lesion in 69 patients, significant stenosis in 303 patients, and vasospasm in 63 patients. Of the 723 patients in the survival outcome group, 273 (168 survivors/105 deaths) underwent PCI. Among the 709 patients in the neurological outcome group, we could confirm the CAG findings in 508 patients: normal coronary, 83 patients; insignificant lesion, 67 patients; significant stenosis, 296 patients; and vasospasm, 62 patients. Among the 709 patients in the neurological outcome group, 268 (144 good neurologic outcomes/124 poor neurologic outcomes) underwent PCI.”
To verify whether our novel nomogram would be applicable to patients irrespective of the CAG results, we performed sensitivity analysis. According to the CAG results, the patients were divided into a group without actual lesions (normal coronary, insignificant lesion) or a group with actual lesions or decreased blood flow (significant stenosis, vasospasm). The sensitivity analysis showed that the AUC values of the novel nomogram for patients with or without actual lesions did not differ significantly from those of our final model in both the survival outcome group and the neurological outcome group. We have added this information in the sensitivity analysis section of the Methods and Results sections, as follows:
2.5. Statistical Analyses
“The predictive accuracy and calibration of the novel nomograms generated with the training set were subsequently tested in the validation set. In addition, to verify whether our novel nomogram would be applicable to patients irrespective of the CAG findings, we performed the sensitivity analysis. According to the CAG findings, the group was divided into a group without actual lesions (normal coronary, insignificant lesion) or a group with actual lesions or decreased blood flow (significant stenosis, vasospasm).”
Results sections
“The sensitivity analysis in patients with or without actual lesions showed that the AUC values of the novel nomogram were 0.8557 (95% CI: 0.7832–0.9281) and 0.9439 (95% CI: 0.8824–0.9999), which were not significantly different to that of the initial nomogram (p=0.53 and p=0.13, respectively).”
Also, we need data related to cardiac markers, echo. etc.
Response: Thank you for your thoughtful comments. In accordance with your comment, we have added troponin I, which had the fewest missing values among all cardiac markers, to Tables 1 and 3. However, echocardiographic findings were not recorded in the KoCARC registry; therefore, we could not report such data.
We added information in the Methods sections and Tables 1 and 3, and we have revised the Limitations section as follows:
Methods sections
The following data were retrieved from the KoCARC registry: patient demographics, comorbidities (hypertension, diabetes mellitus, and dyslipidemia), prehospital characteristics (witnessed arrest, place of arrest, bystander cardiopulmonary resuscitation [CPR], prehospital defibrillation by bystanders or EMS personnel, primary ECG rhythm at the scene, and epinephrine use by EMS providers), hospital characteristics (ROSC at ED arrival, initial ECG rhythm at hospital, and total dose of epinephrine used at hospital), post-ROSC characteristics (E-CPR, TTM, coronary angiographic finding, PCI, and vasopressor use in the hospital), cardiac markers, and time intervals (time from EMS call to scene arrival, time from collapse to ED arrival, time from collapse to ROSC, and time from ED arrival to CAG).
Limitations
“Fourth, this study did not include the data of some variables such as echocardiographic finding, which was important to identify clinically relevant characteristics of the patient."
In addition, I wish the authors shed light on the clinical protocols for E-CPR. Is it standardized among the institutions? EMS protocol pre-hospital and heart cath after ROSC.
Response: In this study, evaluation of E-CPR, CAG, and percutaneous coronary intervention (PCI) after ROSC was performed according to each institution's protocol. E-CPR was performed in only 22 of 62 institutions, and 92 patients in the survival outcome group and 89 patients in the neurological outcome group received E-CPR. E-CPR results have been added to Tables 1 and 3. When our new nomogram was applied to the patient groups with or without E-CPR, the predictive accuracy was not significantly different from that of the initial nomogram (survival outcome group; 0.7850 (0.5856-0.9844), p=0.35 and 0.8886 (0.8389-0.9384), p=0.88, respectively, neurological outcome group; 0.9359 (0.8256-0.9999), p=0.61 and 0.9005 (0.8533-0.9478, p=0.90, respectively). However, this variable was not included in the regression analysis owing to the small number of patients. We have added this information in the Methods sections and have revised the Limitations section as follows:
Methods section
“Patients received cardiopulmonary resuscitation (CPR) in accordance with advanced cardiac life support recommendations; targeted temperature management (TTM), extracorporeal CPR (E-CPR), CAG, and percutaneous coronary intervention (PCI) were conducted according to each institution’s protocol.”
Limitations
“Fifth, in the case of patients who underwent E-CPR, different results were obtained, but the analysis was not conducted owing to the small number of patients who underwent E-CPR in this study.”
Explanation on the Korean EMS system has been added to the Methods section, as follows:
“Korean emergency medical service (EMS) system
The Korean EMS system is a basic-to-intermediate level ambulance service operated by 16 provincial headquarters of the National Fire Department and a single-tiered, fire-based EMS system [11]. The most qualified emergency medical technician (EMT) performs CPR with an automated external defibrillator, evaluates the cardiac rhythm at scene, provides advanced airway management, and administers intravenous fluids. EMTs are not allowed to declare death or stop CPR on the scene unless the patient regains a pulse in the field or during transport to an ED; therefore, all EMS-assessed patients are transported to the nearest ED. In the case of prehospital EMS in Korea, E-CPR is not implemented [12].”
Reviewer 2 Report
The present study is well written, but there are several issues which should be clarified. First, did the data and analysis include both ST-elevation and non-ST elevation on electrocardiography? If the present study included just non-ST elevation, please modify the title of the present study. Second, please provide the data regarding reperfusion therapies. It must be related to in-hospital mortality. Third, in table 2, odds ratios (OR)of age, prehospital defibrillation by EMS providers, and time from collapses to ROSC were different from between survival discharge and good neurologic outcomes, i.e., OR in the former were less than 1.0, but those in the latter were more than 1.0. Please clarify the reasons of these differences.
Minor comment
In table, please align the numbers.
Author Response
Responses to Reviewer #2
The present study is well written, but there are several issues which should be clarified. First, did the data and analysis include both ST-elevation and non-ST elevation on electrocardiography? If the present study included just non-ST elevation, please modify the title of the present study.
Response: We thank you for taking the time to review our manuscript and for providing valuable comments. To clarify, we included both ST-elevation and non-ST elevation. Because the previous version of the manuscript had inappropriate expressions, we have modified the revised manuscript to clearly describe the content in the Methods section.
2.2. Study Population
“The KoCARC registry included data of 874 patients, who underwent CAG regardless of ECG findings (both ST-elevation and non-ST-elevation) after the return of spontaneous circulation (ROSC), registered from October 2015 to June 2018.”
Second, please provide the data regarding reperfusion therapies. It must be related to in-hospital mortality.
Response: Thank you for your thoughtful comments. In accordance with your comment, we have added CAG and PCI data to the Results section, as follows:
“Among the 723 patients in the survival outcome group, we could confirm the CAG findings in 519 patients in the KoCARC registry: normal coronary findings in 84 patients, an insignificant lesion in 69 patients, significant stenosis in 303 patients, and vasospasm in 63 patients. Of the 723 patients in the survival outcome group, 273 (168 survivors/105 deaths) underwent PCI. Among the 709 patients in the neurological outcome group, we could confirm the CAG findings in 508 patients: normal coronary, 83 patients; insignificant lesion, 67 patients; significant stenosis, 296 patients; and vasospasm, 62 patients. Among the 709 patients in the neurological outcome group, 268 (144 good neurologic outcomes/124 poor neurologic outcomes) underwent PCI.”
Third, in table 2, odds ratios (OR)of age, prehospital defibrillation by EMS providers, and time from collapses to ROSC were different from between survival discharge and good neurologic outcomes, i.e., OR in the former were less than 1.0, but those in the latter were more than 1.0. Please clarify the reasons of these differences.
Response: Thank you for the important comment. We have rechecked the statistical analysis section and corrected the incorrect information. The odds ratio has been modified in Table 2 as shown below.
Table 2. Predictors of survival discharge and good neurologic outcomes.
|
Survival discharge |
Good neurologic outcomes |
|||
|
OR (95% CI) |
p-value |
OR (95% CI) |
p-value |
|
|
Age |
0.94 (0.90–0.98) |
<0.001 |
0.95 (0.92–0.98) |
<0.001 |
|
Prehospital defibrillation by EMS providers |
3.52 (1.53–8.05) |
0.003 |
7.84 (3.05–20.17) |
<0.001 |
|
Prehospital ROSC |
2.96 (1.20–7.29) |
0.02 |
|
|
|
Time from collapse to ROSC, min |
0.95 (0.93–0.97) |
<0.001 |
0.91 (0.878–0.94) |
<0.001 |
Minor comment
In table, please align the numbers.
Response: As suggested, we have aligned the numbers in all tables.
Round 2
Reviewer 1 Report
I would like to thank the authors for addressing my comments.
Minor revision: please add percentage next to the numbers of patients with heart catheterization results (page 4) Line 172-179
Author Response
Responses to Reviewer #1
Minor revision: please add percentage next to the numbers of patients with heart catheterization results (page 4) Line 172-179
Response: Thank you for taking the time to review our article and provide thoughtful comments. In accordance with your comment, we have added percentage next to the numbers of patients with heart catheterization results in the Results sections, as follows:
“Among the 723 patients in the survival outcome group, we could confirm the CAG findings in 519 patients in the KoCARC registry: normal coronary findings in 84 patients (16.2%), an insignificant lesion in 69 patients (13.3%), significant stenosis in 303 patients (58.4%), and vasospasm in 63 patients (12.1%). Of the 723 patients in the survival outcome group, 273 (168 survivors/105 deaths) underwent PCI. Among the 709 patients in the neurological outcome group, we could confirm the CAG findings in 508 patients: normal coronary, 83 patients (16.3%); insignificant lesion, 67 patients (13.2%); significant stenosis, 296 patients (58.3%); and vasospasm, 62 patients (12.2%). Among the 709 patients in the neurological outcome group, 268 (144 good neurologic outcomes/124 poor neurologic outcomes) underwent PCI.”